  

# Proteomic analysis reveals microvesicles containing NAMPT as mediators of radioresistance in glioma

Elena Panizza[1], Brandon D Regalado[1], Fangyu Wang[1], Ichiro Nakano[2], Nathaniel M Vacanti[3], Richard A Cerione[1,4], Marc A Antonyak[1]

**Tumor-initiating cells contained within the aggressive brain tumor glioma (glioma stem cells, GSCs) promote radioresistance and disease recurrence. However, mechanisms of resistance are not well understood. Herein, we show that the proteome-level regulation occurring upon radiation treatment of several patient-derived GSC lines predicts their resistance status, whereas glioma transcriptional subtypes do not. We identify a mechanism of radioresistance mediated by the transfer of the metabolic enzyme NAMPT to radiosensitive cells through microvesicles (NAMPT-high MVs) shed by resistant GSCs. NAMPT-high MVs rescue the proliferation of radiosensitive GSCs and fibroblasts upon irradiation, and upon treatment with a radiomimetic drug or low serum, and increase intracellular NAD(H) levels. Finally, we show that the presence of NAMPT within the MVs and its enzymatic activity in recipient cells are necessary to mediate these effects. Collectively, we demonstrate that the proteome of GSCs provides unique information as it predicts the ability of glioma to resist radiation treatment. Furthermore, we establish NAMPT transfer via MVs as a mechanism for rescuing the proliferation of radiosensitive cells upon irradiation.**

## Introduction

Glioma is an aggressive brain cancer with poor clinical outcomes. The standard treatment for the disease involves surgical resection followed by chemotherapy and radiotherapy; however, therapeutic resistance almost invariably arises (Nam & Groot, 2017). Over the last 15 yr, several studies have led to a better understanding of the biology and mechanisms leading to the development and progression of glioma. In 2004, the existence of glioma stem cells (GSCs) was first reported (Singh et al, 2004; Bao et al, 2006). Shortly thereafter, the genomic landscape of glioma was described based on large cohorts of patients (Cancer Genome Atlas Research Network, 2008), facilitating the identification of key genetic and epigenetic alterations, e.g., mutations in isocitrate dehydrogenase 1 and 2 (*IDH1* and *IDH2*) and the glioma-CpG island methylator phenotype (g-CIMP), which are associated with a relatively favorable prognosis (Yan et al, 2009; Noushmehr et al, 2010). In addition, molecular subtyping of glioma was established based on transcriptional signatures (Verhaak et al, 2010). Unfortunately, only two new therapeutic approaches (temozolomide and tumor treating fields) have been approved by the U.S. Food and Drug Administration for the treatment of newly diagnosed glioma in over two decades (Stupp et al, 2005, 2015). Even though a wider range of options exists for treatment of recurrent disease (Shergalis et al, 2018), glioma continues to be a major challenge confronting oncologists, with most patients surviving only 12–18 mo after their initial diagnosis (Nam & Groot, 2017).

Radiotherapy blocks cancer cell proliferation by causing extensive DNA damage. GSCs have generated a good deal of interest after the discovery that they play a causative role in chemotherapy- and radio-resistance (Bao et al, 2006; Skog et al, 2008; Chen et al, 2012). Extracellular vesicles (EVs) released by GSCs have been shown to promote angiogenesis and tumor vascularization (Treps et al, 2017; Lucero et al, 2020); however, their potential role in mediating resistance to radiation is less defined (Ma et al, 2022). EVs are small lipid-enclosed structures that contain bioactive cargo and are classified into two major classes: microvesicles (MVs) that bud off the plasma membrane, and exosomes which are derived from intraluminal vesicles within multivesicular bodies. EVs produced by cancer cells are transferred to other cells within the tumor microenvironment, which may include non-transformed cells and less aggressive cancer cells, and significantly promote their ability to proliferate, and exhibit therapy resistance and invasiveness (Xu et al, 2018; Maacha et al, 2019; Burgos-Ravanal et al, 2021). Cancer cell-derived EVs have also been shown to play important roles in communicating with the surrounding stroma and shaping the tumor microenvironment (Nakano et al, 2015; Desrochers et al, 2016a, 2016b; Wei et al, 2017). Yet, there are still significant gaps in our understanding of the mechanisms by which EVs mediate their effects, particularly as they relate to GSCs.

[1]Department of Molecular Medicine, Cornell University, Ithaca, NY, USA  [2]Department of Neurosurgery, Medical Institute Hokuto Hospital, Hokkaido, Japan  [3]Division of Nutritional Sciences, Cornell University, Ithaca, NY, USA  [4]Department of Chemistry and Chemical Biology, Cornell University, Ithaca, NY, USA

Correspondence: rac1@cornell.edu; maa27@cornell.edu

In the current study, we quantified the proteomic alterations occurring in several patient-derived GSC lines after their irradiation. The analysis enabled us to identify a subset of GSC lines that are resistant to radiation and another one that is radiosensitive. Intriguingly, we find that previous molecular classifications including the *IDH* mutational status and glioma transcriptional subtypes do not predict the GSC radioresistance status. In addition, we identify distinct oncogenic driver proteins that are overexpressed across the resistant GSC lines, suggesting that different disease mechanisms exist among the patients from which the cell lines were derived. We characterize in detail one such mechanism that involves the transfer of the metabolic enzyme nicotinamide phosphoribosyltransferase (NAMPT) to radiosensitive cells, mediated by MVs shed by radioresistant GSCs and glioma cells (NAMPT-high MVs). NAMPT is responsible for regenerating NAD$^+$ from nicotinamide as part of the NAD$^+$ salvage pathway (Cantó et al, 2015; Garten et al, 2015). NAMPT overexpression promotes the progression of glioblastoma, melanoma, colon, breast cancer, and other cancer types (Gujar et al, 2016; Kennedy et al, 2016; Lucena-Cacace et al, 2017, 2018). A way NAMPT contributes to oncogenesis is by providing NAD$^+$ that serves as a cofactor for intracellular enzymes which affect cell survival and responses to DNA damage, including sirtuins (SIRTs) (Luo et al, 2001; Vaziri et al, 2001; Lain et al, 2008), and poly(ADP-ribose) polymerases (PARPs) (Kennedy et al, 2016). Our findings demonstrate that the MV-mediated transfer of NAMPT increases the total NAD(H) level in recipient cells, and promotes their proliferation upon irradiation, exposure to low serum or treatment with the radiation mimetic bleomycin. These effects occur in fibroblasts and in radiosensitive GSCs that were treated with NAMPT-high MVs. We further show that NAMPT transfer and its enzymatic activity in MV-recipient cells are both required to promote the radioresistant phenotype. These findings highlight how some NAMPT-overexpressing GSCs and glioma cells promote resistance to radiation by modulating their surroundings through the shedding and transfer of NAMPT-high MVs. They also raise the interesting possibility that strategies to block the production of MVs or to intervene against the increased NAD(H) levels (and its consequences), can potentially be combined with radiation to more effectively treat glioma patients that overexpress NAMPT.

## Results

### Proteomic profiling identifies a subset of GSC lines that are resistant to radiation

GSCs are thought to be a major source of therapy resistance in glioma (Carruthers et al, 2015; Fidoamore et al, 2016; Garnier et al, 2018; Visvanathan et al, 2018). To characterize mechanisms used by GSCs to promote radiation resistance, we analyzed the proteome of eight GSC lines using liquid chromatography–tandem mass spectrometry (LC–MS/MS). GSC lines were derived from individual patients and either left untreated or treated with six gray of ionizing radiation (Fig S1A), a dose that strongly up-regulates the p21 (CDKN1A)-mediated DNA damage response (Bunz et al, 1998) (Fig S1B). Protein quantification was obtained based on peptide isobaric

labeling, and peptide fractionation by high-resolution isoelectric focusing (HiRIEF) was applied to achieve a comprehensive coverage of the proteome and high quantitative accuracy (Branca et al, 2014; Panizza et al, 2017; Johansson et al, 2019). The analysis identified 120,883 unique peptides corresponding to 9,108 proteins mapping to unique genes (Fig S1A). The examined GSC lines are representative of different glioma molecular subtypes (Verhaak et al, 2010; Verhaak, 2016) (Fig S1C). Analysis of their proteomic response to radiation (Vacanti, 2019) (Table S1) identifies two subsets of GSC lines within treatment groups (Fig 1A). The two subsets display 2,658 and 86 protein levels changing in response to radiation and will be referred to as the radiosensitive and radioresistant groups, respectively (Fig 1B). The relative proliferation of representative radio-sensitive GSC lines was significantly diminished after radiation treatment, whereas the proliferation of radioresistant GSC lines was not affected (Figs 1C and S1D), substantiating the sensitivity status predicted using their proteomic response to radiation. In addition, radiosensitive but not radioresistant GSCs display significant changes in the expression of DNA damage response proteins that are transcriptional targets of p53 (Fischer, 2017). These include the up-regulation of p21 (CDKN1A), which is responsible for cell cycle arrest upon DNA damage (Wade Harper et al, 1993); ribonucleotide–diphosphate reductase subunit M2 B (RRM2B), which is required for DNA synthesis during DNA repair (Tanaka et al, 2000); tumor necrosis factor receptor superfamily member six (FAS), which mediates p53-dependent apoptosis after DNA damage (Müller et al, 1998); and down-regulation of protein aurora borealis (BORA), which is required for progression through mitosis (Seki et al, 2008) (Fig 1D).

We further establish that the GSC transcriptional subtype (Mao et al, 2013) is not associated with their radiosensitivity status (Fig 1A). Similarly, we find that *IDH* mutant-negative GSCs (Baysan et al, 2012; Wang et al, 2017) (Fig S1E) can be either sensitive or resistant to radiation (Fig 1A). This suggests that transcriptional subtyping and *IDH* mutational status do not fully describe resistance to radiation, in line with the concept that additional markers may be necessary to better predict the clinical outcomes of glioma (Sturm et al, 2012; Wang et al, 2017, 2021b; Behnan et al, 2019).

### MVs derived from the radioresistant GSC-267 cell line promote proliferation and are enriched in NAMPT

To pinpoint alterations that are characteristic of radioresistant GSCs, we examined their steady state protein abundances as compared with radiosensitive GSCs (Table S2) and found an enrichment of cell cycle, cell division, and chromosome segregation gene ontology (GO) processes (Fig 2A). This suggests that radioresistant GSCs have an increased ability to progress through the cell cycle and proliferate, which may be beneficial when challenged with radiation-induced DNA damage. Examination of cancer driver and related proteins (CDRPs, Table S3) (Lehtiö et al, 2021a, 2021b) overexpressed in individual radioresistant cell lines (Fig S2A) identifies distinct pathways enriched in each cell line. These include the p53 and the EGFR/PI3K pathways in GSC-1005 cells, and the TGF-beta and the T cell modulation pathway in GSC-267 cells (Fig 2B), indicating that specific mechanisms of resistance distinguish each resistant cell line. Further examination of individual

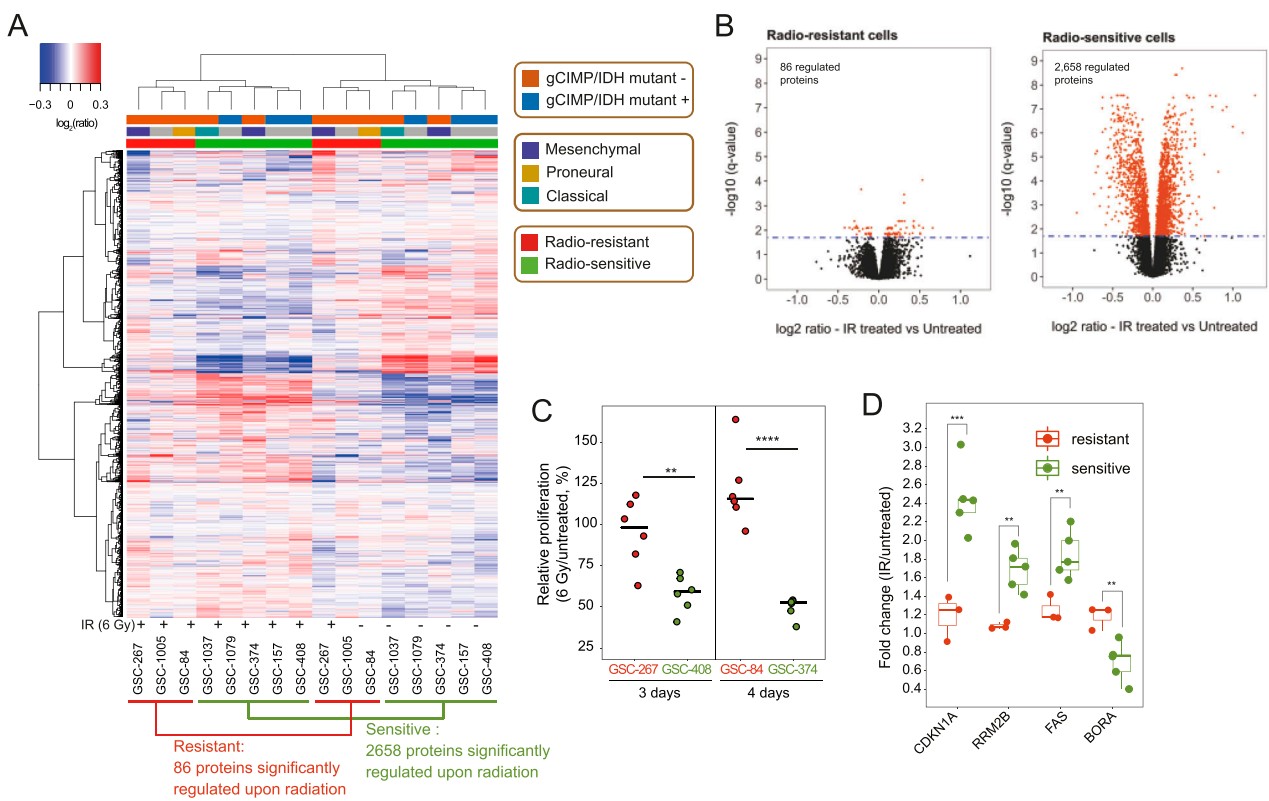

**Figure 1. Proteomic analysis identifies GSCs that are resistant to radiation.**
**(A)** A subset of GSC lines that do not respond to radiation is separated by hierarchical clustering of the quantified proteome based on Euclidian distance and Ward linkage method (Vacanti, 2019). The number of proteins significantly regulated in radioresistant (red) and radiosensitive (green) GSCs upon radiation treatment is displayed. IR, ionizing radiation. Gy, gray; g-CIMP, glioma-CpG island methylator phenotype; *IDH*, isocitrate dehydrogenase. **(B)** Volcano plots representing changes in protein expression upon radiation, plotted against the significance of the regulation. Each dot represents the average $log_2$(IR-treated/untreated) for each protein in the subset of radioresistant (left) or radiosensitive (right) GSC lines. Proteins whose expressions are significantly changed are selected by using the statistical package limma with a Bonferroni–Hochberg corrected q-value lower than 0.02. **(C)** Relative proliferation of representative radioresistant (red) and radiosensitive (green) GSCs after their treatment with 6 Gy of IR. Dots represent independent biological replicates. **(D)** Box plot representing the change in expression of the indicated proteins upon radiation treatment in the examined GSC lines. Red: resistant GSC lines. Green: sensitive GSC lines. Data information: In (C, D), significance levels were evaluated using *t* test. **P-value < $1 \times 10^{-2}$; ***P-value < $1 \times 10^{-3}$; ****, *P*-value < $1 \times 10^{-4}$.
Source data are available for this figure.

cell lines showed that the GO term "vesicle-mediated transport" (Willms et al, 2016) (Fig 2C) is enriched in the radioresistant GSC-267 cells compared with the other GSC lines (Fig S2B). Indeed, electron microscopy images demonstrate the presence of intact vesicles of the expected sizes for MVs and exosomes (Wang et al, 2021a) in the respective preparations isolated from GSC-267 cells (Fig 2D). EVs shed by GSCs are able to enhance tumor angiogenesis (Treps et al, 2017; Lucero et al, 2020); however, their role in altering the response to radiation treatment or other stresses, such as nutrient deprivation, is not well understood. Thus, we examined the effects of MVs and exosomes derived from GSC-267 cells on the proliferation of NIH/3T3 cells cultured in low serum (0.5% calf serum), that is, conditions that typically compromise cell growth and survival. Although exosomes had little effect, MVs significantly increased the proliferation of the recipient cells (Fig 2E). In addition, treatment with a conditioned medium depleted of MVs that was collected from GSC-267 cells did not increase the proliferation of irradiated GSC-408 cells, further suggesting that the MVs are responsible for mediating the effects on proliferation (Fig S2C).

To determine how the MVs mediate this phenotype, we compared their proteomic content with that of exosomes generated by GSC-267 cells using LC–MS/MS. A total of 1,252 proteins were identified as cargo of the EVs (Fig S2D and Table S4). As expected, proteins that are markers for MVs and exosomes (Jeppesen et al, 2019) are elevated in the respective samples (Fig 2F). In addition, cytosolic and nuclear marker proteins (Orre et al, 2019), including the well-established cytosolic and nuclear markers GAPDH, HISTH1B, RPS3, LMNA, and PSMA3, are significantly elevated in whole-cell, but not in EV samples (Figs 2G and S2E). Overall, these observations confirm the identity of MV and exosome fractions isolated from GSC-267 cells and demonstrate that they are devoid of cellular contaminants. To identify the MV protein cargo which is responsible for enhancing the proliferation of recipient cells, we selected proteins that are elevated in GSC-267 cells and in their MVs, but not in their exosomes (66 proteins) (Figs 2H and S2F). Within this set of proteins, a further selection based on their correlation with decreased patient survival highlighted two proteins (Fig S2G), one being the metabolic enzyme NAMPT. NAMPT is

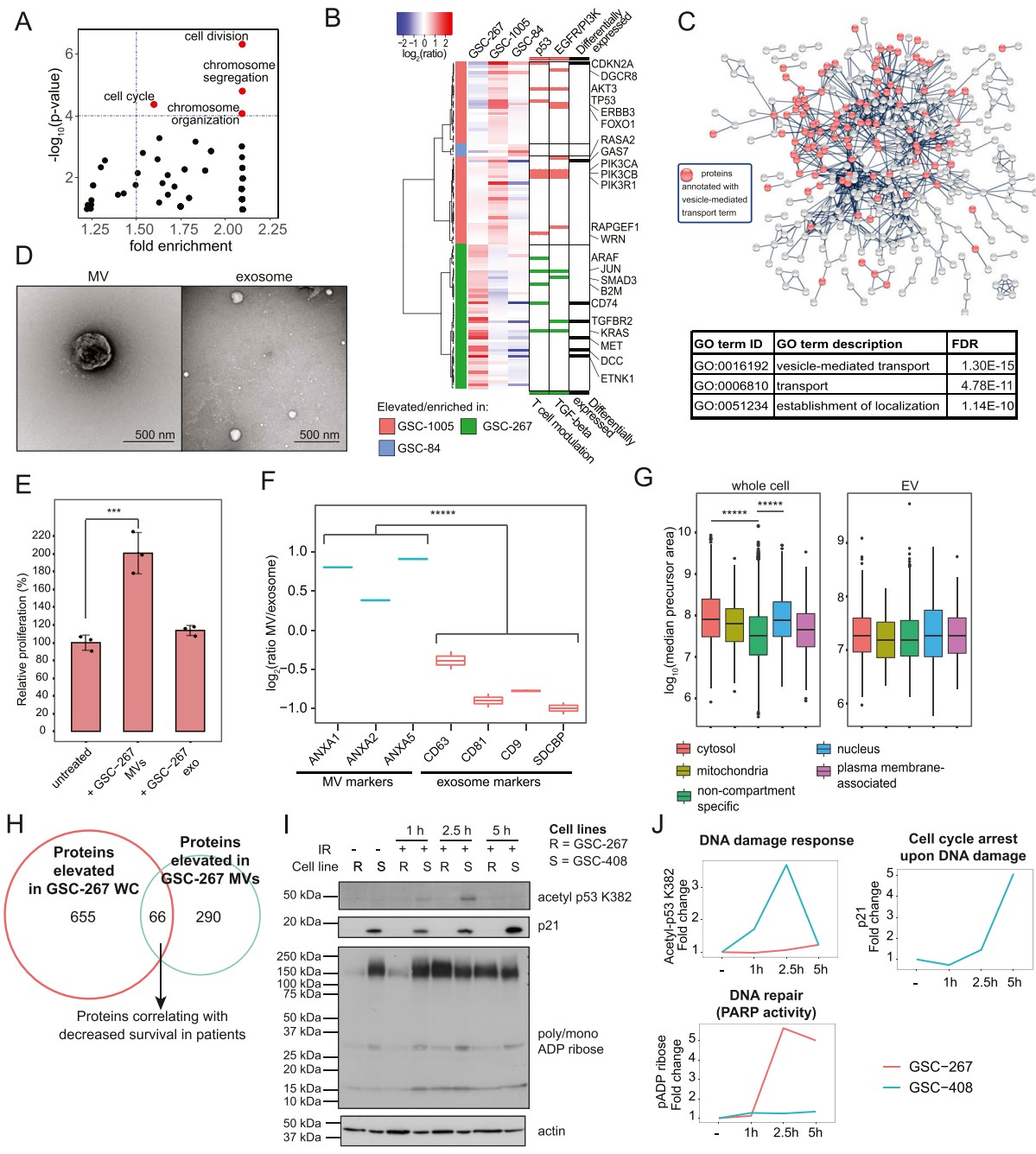

**Figure 2. Microvesicles derived from GSC-267 cells increase the proliferation of recipient cells and are enriched in NAMPT.**
**(A)** GO biological processes enriched in the subset of proteins elevated in radioresistant compared with radiosensitive GSC lines, which were selected using the statistical package limma with a Bonferroni–Hochberg-corrected q-value lower than 0.02 and a log$_2$(ratio) > 0 (n = 81). GO terms "cell division," "cell cycle," "chromosome segregation," and "organization" are significantly enriched (red dots), based on Fisher's exact test P-value lower than 1 × 10$^{-4}$ and fold enrichment higher than 1.5. **(B)** Hierarchical clustering based on Spearman correlation and Ward linkage method of CDRP proteins overexpressed in each of the radioresistant GSC lines compared with the other GSC lines. Significantly overexpressed proteins were selected using the statistical package limma with a Bonferroni–Hochberg-corrected q-value lower than 0.05 and log$_2$(ratio) > 0. Red and green bars in the right panel mark genes belonging to the top two Panther pathways enriched among the CDRPs overexpressed in each cell line. Black bars in the right panel mark top-differentially expressed CDRPs in each cell line (Fig S2A). **(C)** Proteins elevated in GSC-267 cells, compared with other GSC lines, are highly connected in a protein–protein interaction network (protein–protein interaction enrichment P-value < 1 × 10$^{-16}$). The top three GO terms enriched in the network are listed. Vesicle-mediated transport includes the processes of vesicle formation, coating, budding, and fusion with target membranes, and has been shown to describe both MVs and exosomes in a previous proteome-wide analysis (Willms et al, 2016). FDR, false discovery rate-adjusted P-value for the GO terms enrichment, based on Fisher's exact test. **(D)** Transmission electron microscopy images of MVs (left) and exosomes (right) isolated from GSC-267 cells. Scale bar = 500 nm. **(E)** Relative proliferation of NIH/3T3 cells cultured in low serum (0.5% calf serum, CS), either untreated or treated with MVs or exosomes isolated from GSC-267 for 5 d. Individual dots represent independent biological replicates. **(F)** Protein expression levels of EV markers in MVs relative to exosomes isolated from GSC-267 cells (n = 2 for each EV type). ANXA1, 2, 5, Annexin A1, A2, A5; SDCBP, Syntenin-1. **(G)** Distribution of protein abundances based on precursor areas in whole cell (left panel) and EV (right panel) proteomic analyses. Proteins are categorized by subcellular compartments (Orre et al, 2019). The number of compartment-enriched proteins in each category is > 300 in the whole

the rate-limiting enzyme in the NAD$^+$ salvage pathway, responsible for producing NAD$^+$ from nicotinamide (Cantó et al, 2015; Garten et al, 2015). Several studies have shown that the ability of NAMPT to generate NAD$^+$ is important in promoting glioma, and other cancers, by increasing the activity of NAD$^+$-dependent enzymes (Gujar et al, 2016; Kennedy et al, 2016; Lucena-Cacace et al, 2017, 2018). One example is sirtuin-1 (SIRT1), which deacetylates the tumor suppressor p53 on lysine 382 (K382), resulting in its inhibition upon DNA damage (loss of DNA damage response) (Luo et al, 2001; Vaziri et al, 2001; Lain et al, 2008). We indeed show that radiation treatment results in the rapid acetylation of p53 in NAMPT-low GSC-408 cells, but not in NAMPT-high GSC-267 cells. Accordingly, the expression of the p53 transcriptional target p21, which directs cell cycle arrest upon DNA damage, is not detected in GSC-267 cells, whereas it is increased in GSC-408 cells upon radiation (Fig 2I and J). NAD$^+$ is also necessary for the function of PARPs, which initiate DNA repair by adding poly/mono ADP ribose chains (PARylation) at break sites (Kennedy et al, 2016). Indeed, PARylation is strongly activated in GSC-267 cells within 2.5–5 h after irradiation, whereas it is not in GSC-408 cells (Fig 2I and J). Altogether, our results suggest that NAMPT-high cells continue to proliferate upon radiation treatment because of their ability to evade cell cycle arrest (loss of the p53/p21 response) and to better repair damaged DNA (increased PARylation activity).

## NAMPT-high MVs rescue the proliferation of irradiated cells

NAMPT expression levels negatively correlate with patient survival in a number of cancer types (Kennedy et al, 2016), including glioma (Fig 3A). Elevated NAMPT expression also correlates with higher disease grade when examining two distinct patient cohorts where either transcript levels (Fig 3B) or protein levels (Fig 3C) were quantified (Wang et al, 2021b). To specifically assess the role of NAMPT transfer via MVs, we compared the effects of MVs derived from two radioresistant cell lines which overexpress NAMPT, GSC-267, and GSC-84 (Fig S3A). Notably, the expression level of NAMPT does not change after radiation treatment in any of the GSC lines (Fig S3B). Whereas MVs derived from GSC-267 cells contain elevated levels of NAMPT (NAMPT-high MVs), MVs from GSC-84 cells do not contain the enzyme (Fig 3D). Treatment with NAMPT-high MVs, but not with MVs derived from GSC-84 cells, rescues the proliferation of NIH/3T3 cells treated with the radiomimetic drug bleomycin (Povirk, 1996; Zong et al, 2015; Bolzán & Bianchi, 2018) (Fig 3E). We then examined MVs derived from the glioma cell line U-87 MG, which expresses very high levels of NAMPT (Fig S3C) and is able to proliferate when subjected to radiation similar to GSC-267 cells (Fig S3D). MVs derived from U-87 MG cells also contain NAMPT (Fig 3F)

and are able to rescue the proliferation of irradiated NIH/3T3 cells (Fig 3G), overall suggesting that the presence of NAMPT within MVs may be important for promoting radioresistance. We also determined whether NAMPT-high MVs derived from GSC-267 cells are able to rescue the proliferation of less aggressive cancer cells. We show that treatment of the radiosensitive GSC-1079 cells with NAMPT-high MVs restores their ability to proliferate when exposed to radiation (Fig 3H). Finally, we considered whether the MVs could induce a lasting change in recipient cells by conferring long term radioresistance upon them. GSC-408 cells were either left untreated or preconditioned by treating them with MVs derived from GSC-267 cells for 4 d, before being cultured without MVs for an additional 4 d. The cells were then irradiated, and their proliferation determined. We found that radiation treatment decreases the proliferation of the GSC-408 cells that had been preconditioned with MVs to the same extent as for GSC-408 cells that had not been exposed to MVs (Fig S3E), suggesting that continuous exposure to MVs is necessary to maintain radioresistance.

## NAMPT transfer and enzymatic activity are both necessary to rescue the proliferation of MV-recipient cells upon radiation

Treatment of NIH/3T3 cells with NAMPT-high MVs derived from GSC-267 cells results in the transfer of NAMPT protein to the recipient cells, at levels that are proportional to the dose of MVs employed (Fig 4A). Furthermore, NAMPT-high MVs significantly increase the total intracellular NAD(H) level of recipient cells (Fig 4B). To determine whether the presence of NAMPT within MVs is necessary to restore the ability of irradiated cells to proliferate, we generated a GSC-267 cell line (GSC-267 NAMPTsh) where NAMPT expression is knocked down upon addition of doxycycline (Dox), thus markedly reducing the levels of this enzyme in MVs (Figs 4C and D and S4A). Although GSC-267 NAMPTsh cells, –Dox, and +Dox shed similar numbers of EVs (Figs 4E and S4B), the MVs depleted of NAMPT lost the ability to rescue the proliferation of irradiated NIH/3T3 cells (Figs 4F and S4C).

NAMPT promotes cancer progression both through its enzymatic activity, which increases intracellular NAD(H) levels (Kennedy et al, 2016), and by being secreted (extracellular NAMPT, eNAMPT) and binding extracellularly to cell surface receptors such as the C-C chemokine receptor type 5 (CCR5) (Grolla et al, 2016; Jiao et al, 2018; Torretta et al, 2020; Ratnayake et al, 2021). Therefore, we considered whether eNAMPT was associated with the outer surfaces of MVs from GSC-267 cells (Desrochers, et al, 2016; Feng et al, 2017), such that it would be able to engage and activate CCR5 in recipient cells. However, blocking CCR5 with its antagonist cenicriviroc did not inhibit the ability of MVs to rescue the proliferation of NIH/3T3 cells upon bleomycin

cell analysis and > =70 in the EV analysis. **(H)** Identification of proteins that are candidates for promoting the ability of MVs derived from GSC-267 cells to confer aggressive phenotypes. Proteins that are specifically elevated in GSC-267 whole cell (WC) lysates compared with the other GSC lines (n = 721), and proteins that are elevated in MVs generated by GSC-267 cells compared with exosomes (n = 356) are displayed; the overlap between these two sets (n = 66) was selected as candidates. This set of proteins was then filtered based on correlation with decreased survival in a TCGA patient cohort (n = 349). **(I)** Western blots showing the expression levels of acetyl p53 K382, p21, poly/mono ADP ribose and actin in the radioresistant GSC-267 cells (R), and in the radiosensitive GSC-408 cells (S), either left untreated or treated with 6 Gy of ionizing radiation (IR). **(I, J)** Densitometric quantification of protein expression levels displayed in (I). Fold change was calculated relative to the untreated experimental condition for each cell line, and normalized to actin levels. Data information: In (E, F, G), significance levels were evaluated using $t$ test. ***$P$-value < 1 × 10$^{-3}$; *****$P$-value < 1 × 10$^{-6}$.

Source data are available for this figure.

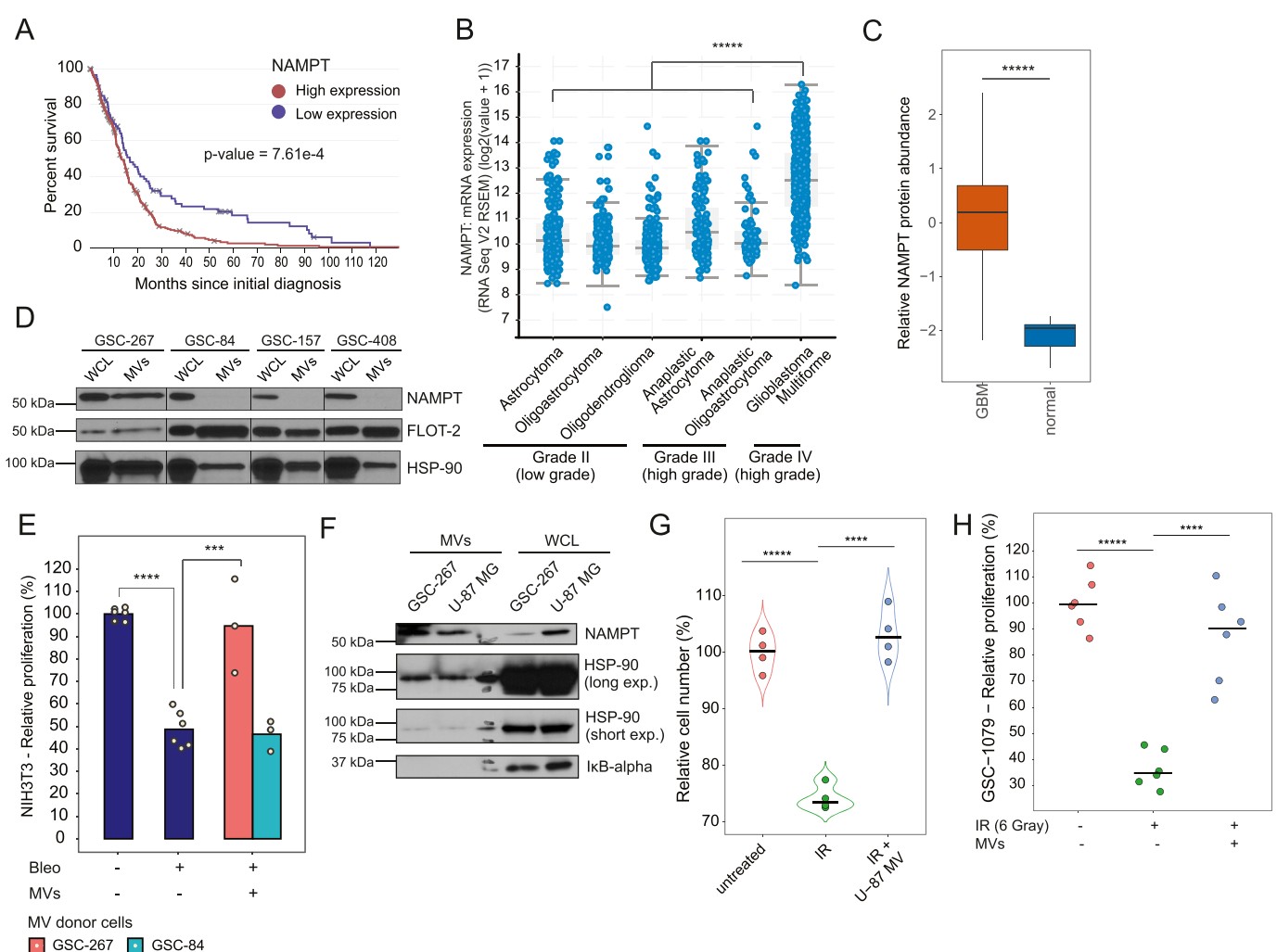

**Figure 3. NAMPT-high MVs mediate radioresistance in recipient cells.**
**(A)** Elevated NAMPT transcript expression significantly correlates with the lower survival of glioblastoma patients (n = 349). The displayed *P*-value was calculated using a logrank test based on data obtained from TCGA. The 25[th] percentile value of NAMPT expression values is set as a threshold to define high expression. **(B)** NAMPT transcript expression levels across glioblastoma (grade IV glioma), high-grade glioma (grade III), and low-grade glioma (grade II). The displayed *P*-value was calculated using a *t* test based on a TCGA cohort of 1,520 patients. **(C)** Relative NAMPT protein abundance in glioblastoma samples (n = 99) compared with normal brain tissue (n = 10) as measured by the National Cancer Institute's Clinical Proteomic Tumor Analysis Consortium (Wang et al, 2021b). **(D)** Western blots showing the expression levels of NAMPT and the MV markers flotillin-2 (FLOT-2) and heat shock protein 90 (HSP-90) in GSC lines and in the MVs derived from these cells. Whole cell lysates (WCL), whole cell lysate. **(E)** Relative proliferation of NIH/3T3 cells cultured in low serum (0.5% CS) medium, either left untreated or treated with the indicated combinations of 5 $\mu$M bleomycin (Bleo) and MVs derived from GSC-267 or GSC-84 cells for 5 d. Individual dots represent independent biological replicates. **(F)** Western blots representing the expression levels of NAMPT in GSC-267 and U-87 MG WCL and MVs. HSP-90 is used as an EV marker, whereas IKB-alpha is a WCL marker. **(G)** Relative number of viable NIH/3T3 cells cultured in low serum (0.5% CS) and treated with the indicated combinations of 6 Gy of ionizing radiation (IR) and MVs derived from U-87 MG cells. Cells were counted after 4 d. Individual dots represent independent biological replicates. **(H)** Relative proliferation of radio-sensitive GSC-1079 cells either untreated or treated with 6 Gy of ionizing radiation (IR) alone or in combination with MVs derived from GSC-267 cells for 4 d. Individual dots represent independent biological replicates. Data information: in (B, C, E, G, H), significance levels were evaluated using *t* test. ***P*-value < 1 × 10$^{-3}$; ****P*-value < 1 × 10$^{-4}$; *****P*-value < 1 × 10$^{-6}$.
Source data are available for this figure.

treatment (Fig S4D). Furthermore, addition of human recombinant NAMPT, which can bind and activate CCR5 (Yoshida et al, 2019; Ratnayake et al, 2021), fails to rescue the proliferation of irradiated fibroblasts and radiosensitive GSC-408 cells (Figs 4G and S4E). Conversely, treatment of irradiated GSC-408 cells with nicotinamide mononucleotide (NMN), the enzymatic product of NAMPT, rescues their proliferation (Fig 4G). Similarly, treatment of NIH/3T3 cells cultured in low serum with MVs or with NMN increases their proliferation, whereas the combination of MVs and

NMN does not provide an additional proliferative advantage. Moreover, treatment with the NAMPT enzymatic inhibitor FK-866 prevents the MV-mediated increase in proliferation, but it does not affect the proliferation of NIH-3T3 cells that were also treated with NMN (Fig 4H). Collectively, these findings demonstrate that the ability of NAMPT-high MVs to rescue the proliferation of recipient cells exposed to radio-mimetic treatment, low serum, and radiation treatment is dependent on the transfer and enzymatic activity of NAMPT in recipient cells.

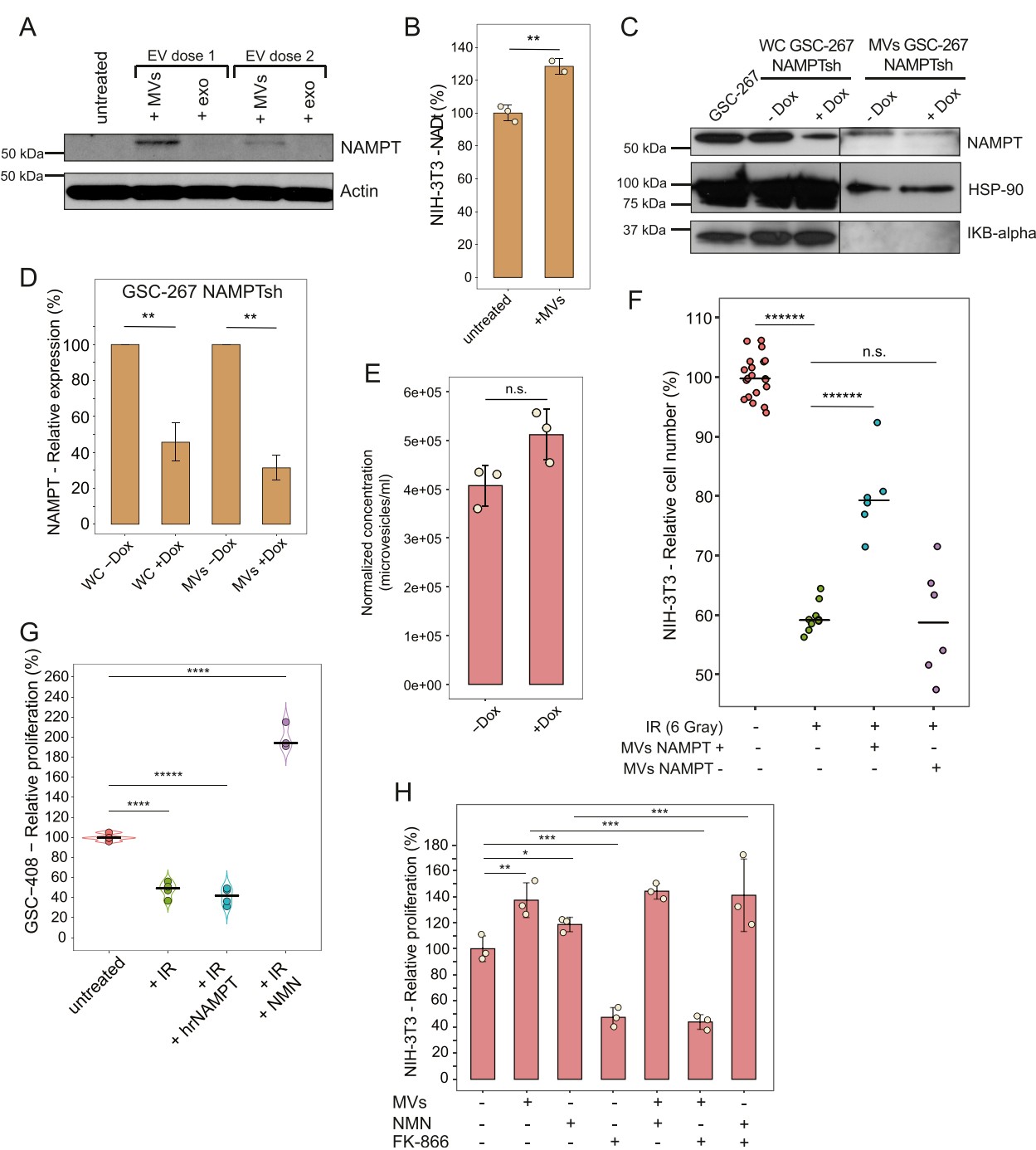

**Figure 4. NAMPT enzymatic activity and transfer are required to promote the proliferation of irradiated cells.**
**(A)** Western blots representing the expression levels of NAMPT in NIH/3T3 cells, either untreated or treated with progressively lower doses of MVs and exosomes (exo) isolated from GSC-267 cells (EV dose 2 is half of EV dose 1) for 24 h. Actin was used as a loading control. **(B)** Total NAD⁺ and NADH level (NADt) in NIH/3T3, untreated or treated with MVs derived from GSC-267 cells for 6 h. Individual dots represent independent biological replicates. Significance of observed changes was evaluated using Student's $t$ test. **$P$-value $< 1 \times 10^{-2}$. **(C)** NAMPT protein expression levels in whole cell (WC) lysates from GSC-267 parental cells, GSC-267 NAMPTsh cells −/+ Dox, and in MV protein extracts derived from GSC-267 NAMPTsh cells −/+ Dox. IKB-alpha is a WC marker and shows that the MV preparations are devoid of cellular contaminants. HSP-90 was used as a loading control. **(C, D)** Densitometric quantification of NAMPT protein expression levels displayed in (C). Variability is evaluated based on two technical replicates. Significance of observed changes was evaluated using $t$ test. **$P$-value $< 1 \times 10^{-2}$. **(E)** Concentration of MVs (vesicles larger than 200 nm) isolated from GSC-267 NAMPTsh cells −/+Dox, normalized based on cell number. Individual dots represent independent biological replicates. The significance of observed changes was evaluated using $t$ test. n.s., not significant. **(F)** Relative number of viable NIH/3T3 cells cultured in low serum (0.5% CS) and treated with the indicated combinations of 6 Gy of ionizing radiation (IR) and MVs derived from GSC-267 NAMPTsh cells −Dox (MVs NAMPT +) or + Dox (MVs NAMPT −) for 4 d. Individual dots represent independent

## Discussion

GSCs are critical drivers of resistance to radiation and chemotherapy (Bao et al, 2006; Chen et al, 2012). Cultures of GSCs derived from individual patients reliably recapitulate the cellular diversity of the original tumors (Mao et al, 2013). Although previous studies have identified targets that could sensitize GSCs to radiation (Bao et al, 2006; Bhat et al, 2013; Visvanathan et al, 2018; Shi et al, 2021), so far, none have progressed to the clinic. Here, we present a comprehensive proteomic profile of a set of patient-derived GSC lines in response to radiation and introduce a previously undescribed mechanism for radio-resistance that involves the MV-mediated transfer of the metabolic enzyme NAMPT to recipient cells. Recently, the proteomic landscape of glioma was described by the Clinical Proteomic Tumor Analysis Consortium (CPTAC), based on a collection of 99 treatment-naïve tumor samples (Wang et al, 2021b). In the CPTAC study, proteome-level information captured a subset of patients that belong to a different subtype compared with previously defined transcriptional subtypes. More broadly, studies that employed proteomics to characterize the clinical landscape of cancer have generated insights into immune-based subtyping, oncogenic drivers, and patient stratification, which could not be captured by genomic or transcriptomic analyses alone (Clark et al, 2019; Johansson et al, 2019; Dou et al, 2020; Hu et al, 2020; Huang et al, 2021). In this study, we identify distinct oncogenic driver proteins that are overexpressed in different radioresistant patient-derived GSCs (Fig 2B). Furthermore, we demonstrate that proteome-level changes accurately predict glioma radiosensitivity status, whereas transcriptional subtypes or *IDH*-mutant status do not (Fig 1A). Our study thus confirms that the proteome provides unique information that is important to understanding disease progression.

The characterization and definition of EVs is an active field of research and many recent advances have been made (reviewed in Hur et al [2020]; Panizza et al [2020]; van Niel et al [2018]). A relatively small number of studies have used systems biology approaches that broadly characterize the transcriptomic and proteomic content of EVs, and most studies still rely on the detection of a few specific markers (Wang et al, 2021a). Here, we present a comprehensive overview of the proteome of EVs generated by GSCs. The analysis identified over 1,200 proteins contained within EVs (Fig S2D). The large number of proteins detected allowed for unbiased target discovery (Fig 2H), and the comprehensive proteome of GSC-derived EVs presented here can be used as a resource for future studies (Table S4). Well-established EV markers were reliably detected within MVs and exosomes using proteomics (Fig 2F). Loss of enrichment in intracellular proteins was used to effectively demonstrate the lack of cellular contaminants in EV preparations (Figs 2G and S2E), thus demonstrating that systems biology approaches can be applied to determine the purity and identity of EV preparations.

Herein, we demonstrate that NAMPT is a biologically relevant cargo of MVs produced by NAMPT-overexpressing and radio-resistant GSCs and glioma cells (Fig 3D and F). Although the existence of EVs containing NAMPT has been reported before (Yoshida et al, 2019; Ratnayake et al, 2021), our findings demonstrate that they are able to enhance the proliferation of recipient cells that were irradiated (Figs 3G and H and 4F) or exposed to other types of cytotoxic stresses (Figs 2E, 3E, 4G, and S4D). Therefore, our study provides an important contribution to explaining how NAMPT overexpression enhances tumor overall resistance to therapy, aggressive phenotype, and decreases patient survival (Sampath et al, 2015; Kennedy et al, 2016; Lucena-Cacace et al, 2018; Yaku et al, 2018). The extent to which the EV-mediated transfer of NAMPT occurs in the tumor microenvironment remains to be determined, and a possible way to examine it will be to quantify differences in NAMPT protein expression levels across single cells within tumor sections. This is an important question that will be worth addressing in future studies.

Clinical trials that tested compounds targeting NAMPT have failed because of toxicity (Hovstadius et al, 2002; Ravaud et al, 2005; Holen et al, 2008; Pishvaian et al, 2009; Von Heideman et al, 2010; Goldinger et al, 2016), primarily because NAMPT is essential for normal cell metabolism (Garten et al, 2015). However, the important contribution of NAMPT to cancer progression warrants the examination of alternative strategies. An avenue to treating NAMPT-overexpressing tumors may be to identify targets that rely on NAD(H), because NAMPT overexpression increases intracellular NAD(H) levels (Garten et al, 2015). Examples include the PARP family of enzymes, which consume NAD$^+$ to facilitate DNA damage repair (Kennedy et al, 2016); and autophagy, which is activated by NAD$^+$ and protects cells from radiation-induced DNA damage (Zhu et al, 2016; Dolgin, 2019). Importantly, additional mechanisms of radioresistance that rely on NAD(H) have not been extensively studied. Therefore, their identification may provide the key to treating glioma cases where NAMPT is overexpressed. Moreover, because NAMPT can be detected directly in plasma and cerebrospinal fluid samples (Hallschmid et al, 2009; Hara et al, 2011; Audrito et al, 2015; Grolla et al, 2015; Moon et al, 2016; Welton et al, 2017; Macron et al, 2020), it might serve as a valuable marker for prognosis and selecting patients for therapy.

## Materials and Methods

### Cell culture and treatment

GSC-267, GSC-1005, GSC-84, GSC-1037, GSC-1079, GSC-374, GSC-157, and GSC-408 cell lines were generated from individual glioma tumor samples that were freshly resected from human patients. GSCs were established as de-identified, permanent cell lines in the

---

biological replicates. Significance of observed changes was evaluated using *t* test. *****P*-value < 1 × 10$^{-6}$. **(G)** Relative proliferation of GSC-408 cells treated with the indicated combinations of 6 Gy of ionizing radiation (IR), 19 nM human recombinant NAMPT (hrNAMPT), and 500 μM nicotinamide mononucleotide for 4 d. Individual dots represent independent biological replicates. **(H)** Relative proliferation of NIH/3T3 cells cultured in low serum (0.5% CS) and treated with the indicated combinations of MVs derived from GSC-267 cells, 500 μM nicotinamide mononucleotide, and 10 μM of the NAMPT inhibitor FK-866 for 3 d. Individual dots represent independent biological replicates. Significance of the observed changes was evaluated using *t* test. *P*-value < 0.02; **P*-value < 1 × 10$^{-2}$; ***P*-value < 1 × 10$^{-3}$. Data information: in (B, D, E, F, G, H), significance levels were evaluated using *t* test. *P*-value < 0.02; **P*-value < 1 × 10$^{-2}$; ***P*-value < 1 × 10$^{-3}$; ****P*-value < 1 × 10$^{-4}$; *****P*-value < 1 × 10$^{-5}$; ******P*-value < 1 × 10$^{-6}$.
Source data are available for this figure.

laboratory of Dr. Nakano as previously described (Bhat et al, 2013; Gu et al, 2013; Mao et al, 2013). GSC lines were characterized based on mRNA profiling and/or immunostaining, as described upon their publications (Guvenc et al, 2013; Mao et al, 2013; Kim et al, 2016; Adnani et al, 2022; Alhalabi et al, 2022). GSC identity was also confirmed based on their expression of signatures corresponding to mesenchymal, classical or proneural subtypes (Verhaak et al, 2010), as detailed in Fig S1C. The cells were cultured in Dulbecco's Modified Eagle Medium:Nutrient Mixture F-12 (DMEM/F-12) (cat. no. 21331-020; Gibco) with the addition of B-27 supplement to a final concentration of 2% vol/vol (cat. no. 17504044; Gibco), heparin 2.5 mg/ml (cat. no. H3149; Sigma-Aldrich), basic fibroblast growth factor (bFGF, 20 ng/ml) (cat. no. 100-18B; Pepro Tech. Inc.), epidermal growth factor (EGF, 20 ng/ml) (cat. no. AF-100-15; Pepro Tech. Inc.), and penicillin and streptomycin (cat. no. 15140122; Gibco) to a final concentration of 100 IU/ml and 100 μg/ml, respectively. GSC spheres were dissociated mechanically by pipetting or using TrypLE Express Enzyme (cat. no. 12605028; Thermo Fisher Scientific), and bFGF and EGF were added twice a week. For proteomic analysis, GSCs were seeded in 175 cm$^2$ flasks using fresh complete medium at a confluency of about 50–60% and a volume of 50 ml 2 d before radiation treatment. GSCs were either left untreated or administered a single dose of 6 gray of ionizing radiation using a Mark I Model 68 Cesium-137 gamma irradiator at the Irradiator Facility within the Transgenic Mouse Core Facility at Cornell University. The cells were harvested for proteomic analysis 2 d after irradiation. Murine NIH/3T3 cells (ATCC CRL-1658) were cultured in DMEM media (cat. no. 11965092; Gibco) supplemented with 10% (vol/vol) calf serum (cat. no. 26010074; Gibco) and penicillin and streptomycin to a final concentration of 100 IU/ml and 100 μg/ml, respectively. Human embryonic kidney (HEK)-293T (ATCC CRL-11268) and U-87 MG (ATCC HTB-14) cells were cultured in DMEM media supplemented with 10% (vol/vol) cat. no. 10437028; FBS and penicillin and streptomycin to a final concentration of 100 IU/ml and 100 μg/ml, respectively. Compounds used for treatment include the following: bleomycin sulfate (cat. no. 13877; Cayman Chemicals); cenicriviroc, (cat. no. S8512; Selleck Chemicals); β-Nicotinamide mononucleotide (NMN), ≥95% HPLC (cat. no. N3501; Sigma-Aldrich, Millipore Sigma); Recombinant Human Visfatin (cat. no. 130-09; PeproTech); FK-866 (cat. no. 13287; Cayman Chemical Company). When EVs were used to treat cells, equal amounts were added to each experimental well. EV amounts were normalized based on number of donor cells: EVs derived from 125,000 EV donor cells were used to treat individual 96-well plate wells containing 2,000 recipient cells and were added every 24 h.

### Protein extraction

GSCs were transferred to conical tubes and pelleted by centrifugation at 100*g*. The cells were washed twice with PBS (cat. no. 14190250; Life Technologies) and dry cell pellets were stored at −80°C. The cells were lysed in the presence of 4% (wt/vol) SDS (cat. no. 71725; Sigma-Aldrich), 25 mM HEPES (cat. no. H3375; Millipore Sigma), pH 7.6, and 1 mM DTT (cat. no. D11000; Research Products International) (4% SDS cell lysis buffer). Adherent cells were scraped off the culture plates in the presence of 4% SDS cell lysis buffer. Lysates were then heated at 95°C for 10 min,

sonicated, and then centrifuged for 15 min at 14,000*g* and 4°C. Supernatants were transferred to new vials, and protein concentrations were quantified using the DC protein assay (cat. no. 5000112; Bio-Rad).

### Protein digestion for whole cell proteomic analyses

Protein extracts were processed after the Single-pot, solid-phase-enhanced sample preparation (SP3) protocol (Hughes et al, 2014; Sielaff et al, 2017) with slight modifications. Briefly, a bead slurry was prepared by mixing 50 μl of Sera-Mag SpeedBeads Carboxyl Magnetic Beads (hydrophobic, cat. no. 10204-670; Cytiva, VWR) with 50 μl of Sera-Mag SpeedBeads Carboxyl Magnetic Beads (hydrophilic, cat. no. 10204-628; Cytiva, VWR). The bead slurry was washed two times with 200 μl of MQ water using a magnetic rack, before resuspension in 500 μl of MQ water. Protein samples (200 μg each) were diluted to a final volume of 200 μl using the cell lysis buffer described above. Cysteine residues were alkylated by adding 40 μl of SP3 bead slurry to the protein samples, and chloroacetamide (cat. no. 10204-670; Millipore Sigma) to a final concentration of 40 mM, and acetonitrile (ACN) (cat. no. 10204-670; Millipore Sigma) to a final concentration of 70% (to bind proteins to the SP3 beads). Samples were then incubated for 20 min on a rotating rack. After the incubation, supernatants were removed and SP3 beads were washed twice with 200 μl of 70% EtOH and once with 180 μl of 100% ACN using a magnetic rack. SP3 beads were air-dried for 30 s before digestion at 37°C overnight in 100 μl of a solution containing 50 mM HEPES pH 7.6, 1 M urea, and 4 μg of Lysyl Endopeptidase (Lys-C) (cat. no. 129-02541; Wako Chemicals) with mild shaking. Subsequently, 100 μl of a solution containing 50 mM HEPES pH 7.6 and 4 μg trypsin (cat. no. 90057; Pierce) was added and samples were incubated again at 37°C overnight with mild shaking. Supernatants were transferred to new tubes and peptide concentrations were quantified using the DC Protein Assay. Aliquots corresponding to 50 μg of each sample were set aside for TMT labeling. Identical linker samples were prepared to function as a denominator in each TMT set. Linker samples were prepared by pooling equal amounts of proteins from each of the 16 samples, up to a final amount of 50 μg each. All samples were lyophilized before TMT labeling.

### TMT labelling

Peptide samples were labeled with 10-plex TMT reagents (cat. no. 90110; Thermo Fisher Scientific) as previously described (Panizza et al, 2017). Briefly, before labeling samples were resuspended using TEAB pH 8.5 (50 mM final concentration) (cat. no. T7408; Millipore Sigma) to adjust the pH. Each sample was labeled with an isobaric TMT-tag. Labelling efficiency was verified by LC–MS/MS before pooling the samples. Pooled samples were desalted with reversed phase-solid phase extraction cartridges (cat. no. 8B-S001-DAK; Phenomenex) and then lyophilized in a SpeedVac (Thermo Fisher Scientific). Sample clean-up was performed by solid phase extraction (cat. no. 8B-S029-AAK; SPE strata-X-C, Phenomenex). Purified samples were dried in a SpeedVac.

## Peptide-level HiRIEF

HiRIEF was performed as previously described (Branca et al, 2014). Briefly, peptides from whole-cell samples were focused on immobilized pH gradient (IPG) gel strips on a linear pH range of 3-10 (cat. no. 17-6002-44; Cytiva). Strips were divided into 72 fractions and extracted to V-bottom 96-well plates with a liquid handling robot (GE Healthcare prototype modified from Gilson Liquid Handler 215). Plates were lyophilized in a SpeedVac before LC–MS/MS analysis. Dried peptides were dissolved in 3% ACN/0.1% formic acid (FA) (cat. no. 00940; Millipore Sigma) and consolidated into 40 fractions based on fraction complexity. Specifically, less complex fractions were pooled: that is, fractions 19-26, 31-35, 42-49, 53-63, 66-70.

## EV isolation

EVs were isolated as described before (Wang et al, 2021a). Briefly, a medium containing GSCs was subjected to two consecutive centrifugations at 100*g* to clarify the medium of cells and debris. The partially clarified medium was filtered using a 0.22 *µm* pore size Steriflip PVDF filter (cat. no. SE1M179M6; Millipore Sigma). MVs were collected off the filter by adding 4% SDS cell lysis buffer for protein extraction or cell culture medium for functional assays. The filtrate was subjected to ultracentrifugation at 100,000*g* for 3 h to collect exosomes. The exosome pellet was either collected by adding 4% SDS cell lysis buffer for protein extraction or cell culture medium for functional assays.

## Protein extraction, digestion, and TMT labeling for EV proteomic analysis

GSC-267 cells were seeded in seven 175 cm$^2$ flasks using fresh complete medium at a confluency of about 60–70% and a volume of 50 ml the day before EV collection. MVs and exosomes were isolated as detailed above, and proteins were extracted using 130 *µ*l of 4% SDS lysis buffer. Protein digestion was performed using the SP3 protocol described above for whole cell samples, but volumes were adapted as follows. EV protein samples (~10–40 *µ*g each) were diluted to a final volume of 120 *µ*l using 4% SDS cell lysis buffer. Cysteine residues were alkylated by adding chloroacetamide to a final concentration of 40 mM, 6 *µ*l of SP3 bead slurry, and ACN to a final concentration of 70% (to bind proteins to the SP3 beads). Samples were then incubated for 20 min on a rotating rack. After the incubation, supernatants were removed and SP3 beads were washed twice with 100 *µ*l of 70% EtOH and once with 100 *µ*l of 100% ACN using a magnetic rack. SP3 beads were air-dried for 30 s before digestion at 37°C overnight in 20 *µ*l of a solution containing 50 mM HEPES pH 7.6, 1 M urea and 0.6 *µ*g of Lysyl Endopeptidase with mild shaking. Subsequently, 20 *µ*l of a solution containing 50 mM HEPES pH 7.6 and 0.6 *µ*g trypsin was added and samples were incubated again at 37°C overnight with mild shaking. Supernatants were transferred to new tubes and peptide concentrations were quantified using the DC Protein Assay. Peptides were labeled using 10-plex TMT reagents as described above and 4 *µ*g of proteins were used for each TMT channel. Linker samples were prepared by pooling four EV samples, corresponding to a final amount of 4 *µ*g of peptides.

## LC–MS/MS

LC–MS/MS analysis was performed by the Clinical Proteomics Mass Spectrometry facility at Karolinska Institutet-Karolinska University Hospital, Science for Life Laboratory in Stockholm, Sweden. Peptide samples were separated using a reversed-phase gradient containing phase A solution (5% dimethyl sulfoxide, DMSO, 0.1% FA) (cat. no. 472301, DMSO; Millipore Sigma) and phase B solution (90% ACN, 5% DMSO, 5% water, and 0.1% FA). For whole cell proteomic analysis, each HiRIEF fraction was analyzed independently using a gradient that proceeded from 6% phase B to 37% phase B over 30–90 min depending on the complexity of the fraction being analyzed. For EV proteomic analysis, peptides were separated using a gradient that proceeded from 6% phase B to 30% phase B over 180 min. Upon completion of the gradient, the column was washed with a solution of 99% phase B for 10 min and re-equilibrated to the initial composition. A nano EASY-Spray column (PepMap Rapid Separation Liquid Chromatography; C18; 2-*µ*m bead size; 100 Å pore size; 75-*µ*m internal diameter; 50 cm long; Thermo Fisher Scientific) was used on the nanoelectrospray ionization EASY-Spray source at 60°C. Online LC–MS/MS was performed using a hybrid Q Exactive mass spectrometer (Thermo Fisher Scientific). Fourier transform–based mass spectrometer master scans with a resolution of 60,000 (and mass range 300–1,500 m/z) were followed by data-dependent MS/MS (35,000 resolution) on the 5 most abundant ions using higher-energy collision dissociation at 30% normalized collision energy. Precursor ions were isolated with a 2 m/z window. Automatic gain control targets were 1 × 10$^6$ for MS1 and 1 × 10$^5$ for MS2. Maximum injection times were 100 ms for MS1 and 100 ms for MS2. The entire duty cycle lasted ~1.5 s. Automated precursor ion dynamic exclusion was used with a 60-s duration. Precursor ions with unassigned charge states or a charge state of +1 were excluded. An underfill ratio of 1% was applied.

## Proteomics database search and ratio calculation

Raw MS/MS files for the whole cell proteomics analysis were converted to mzML format using msConvert from the ProteoWizard tool suite (v3.0.19127) (Holman et al, 2014). Spectra were then searched using MSGF+ (v2018-07-21) and Percolator (v3.01) (Granholm et al, 2014) using the Galaxy platform (Boekel et al, 2015). The reference database was the *Homo sapiens* protein subset of Swiss-Prot, canonical isoforms, released on 2018-08-02. MSGF+ settings included precursor ion mass tolerance of 10 ppm and peptide spectral matches (PSMs) allowed for up to two missed trypsin cleavages. Carbamidomethylation on cysteine and TMT 10-plex on lysine and the N terminus were set as fixed modifications, and oxidation of methionine was set as a dynamic modification while searching all MS/MS spectra. Label-free quantification was calculated as the median MS1 precursor area across the two TMT sets for each gene (Table S2). Quantification of TMT 10-plex reporter ions was performed using OpenMS project's IsobaricAnalyzer (v2.0) (Sturm et al, 2008). A false discovery rate cutoff of 1% was applied at the PSM level. To obtain relative quantification of steady state protein abundances for whole-cell samples, ratios were first normalized to the median of each TMT channel, assuming equal peptide loading of all samples. Then, the TMT reporter ion value for

each PSM was divided by the TMT reporter ion value for channel 131 (the linker) in each TMT set to normalize protein quantification across the two TMT sets. Finally, the median ratio of PSMs belonging to a unique gene symbol was used to obtain gene-centric protein quantifications. TMT ratios were then $\log_2$ normalized (Table S2). To obtain relative quantification of protein level changes upon radiation, protein TMT ratios for each protein were subtracted with the average of untreated and treated for each cell line. This normalization method allows to highlight protein expression changes upon treatment (Table S1). MS/MS spectra for the EV proteomic analysis were matched against the human and bovine subset of the Swiss-Prot database, release 2018-08-02, using Sequest/Percolator under the Proteome Discoverer software platform (PD 1.4, Thermo Fisher Scientific). Settings for the search were the same as specified before (Panizza et al, 2017). Peptides that matched to the bovine reference proteome were filtered out. Label-free quantification was performed as described before (Pernemalm et al, 2019). Briefly, protein MS1 precursor areas were calculated as the average of its top three most intense peptide precursor areas for each TMT set. Then, the median of the two TMT sets was used as MS1 precursor area. Inferred gene identity false discovery rates were calculated using the picked-FDR method. To calculate TMT ratios, first, the TMT reporter ion value for each PSM was divided by the TMT reporter ion value for channel 128C (the linker in set 1) or 130C (the linker in set 2) to normalize protein quantification across the two TMT sets. Then, TMT ratios were $\log_2$ normalized and the average of the four EV samples (two MV, two exosome samples) for GSC-267 cells was subtracted to each ratio for each individual protein to obtain relative protein abundances in MV compared with exosomes (Table S4).

### g-CIMP/*IDH* mutational status and transcriptional subtyping

g-CIMP+/*IDH* mutant status was identified based on an established 100 gene signature (Baysan et al, 2012; Wang et al, 2017). Transcriptional subtypes for the five *IDH* WT GSC lines were established based on published gene signatures (Verhaak et al, 2010). For both analyses, gene set enrichment analysis (GSEA) scores were calculated using the R package GSVA (Hänzelmann et al, 2013) specifying "ssgsea" as method. To assign samples to a mutational status or to a subtype, a threshold score was defined based on GSEA scores for a set of 99 glioblastoma samples (Wang et al, 2021b) with a known mutational status and molecular subtype.

### Data visualization, bioinformatics analyses, and statistics

All plots were generated using RStudio. The statistical package linear models for microarray data (limma) (Ritchie et al, 2015) were used to define significantly regulated proteins for all proteomic data analyses. GO enrichment analysis was performed using the R package TopGO (Alexa & Rahnenfuhrer, 2021) and Fisher's exact test was employed to evaluate the significance of the enrichment. Protein–protein interaction analysis was performed using the web tool provided by the Search Tool for the Retrieval of Interacting Genes/Proteins (STRING; https://string-db.org) (Szklarczyk et al, 2017). High confidence interactions (interaction score > 0.900) were considered for the analysis. Patient survival was analyzed in

relation to NAMPT transcript expression based on data generated by TCGA using the Affymetrix platform HT HG U133A (Cancer Genome Atlas Research Network, 2008; Brennan et al, 2013), through the web tool available at https://www.betastasis.com. CDRPs, 833 genes, Table S3 were defined as previously described (Lehtiö et al, 2021a, 2021b). Significant overexpression of CDRPs in each radio-resistant GSC line was assessed by comparing steady-state protein abundances for the two replicates (untreated and treated) for each cell line with protein abundances in all the other cell lines. Panther pathways (Mi et al, 2021) enriched within the set of CDRPs overexpressed in each resistant GSC line were filtered based on the following criteria: > 200 proteins the reference list; > 3 proteins in the input list; *P*-value < 0.005. All applied statistical analyses and number of biological replicates are specified in the figures and corresponding figure legends. Experiments were repeated at least two times unless stated otherwise.

### Cell proliferation assay

Relative cellular proliferation was measured based on the Cell Counting Kit-8 (CCK-8) assay (cat. no. CK04; Dojindo Molecular Technologies, Inc.), following the manufacturer's instruction. Briefly, cells were seeded in 96-well plates and treated for the indicated amount of time before adding a volume of 10 $\mu$l of CCK-8 reagent. The reaction was incubated for 2–6 h before reading absorbance using a plate reader. The absorbance values of cell culture wells that contained only cell culture medium was used as a background and subtracted from the absorbance values obtained from the experimental wells. Relative proliferation was calculated as a percentage.

### Cell counting assay

Number of cells was counted using an imaging cytometer (Celigo, Nexcelom Bioscience) as previously described (Blum et al, 2021). Briefly, cells were seeded in 96-well plates, black, clear bottom (cat. no. 655090; Greiner), and stained using Hoechst dye (cat. no. H3570; Life Technologies) and propidium iodide (cat. no. P3566; Thermo Fisher Scientific) to measure total and dead cell number. The number of viable cells, obtained by subtracting dead from total cell number, is reported.

### Assay for viable cell counting based on intracellular protease content

Number of viable cells was counted using the CytoTox-Glo Cytotoxicity Assay (Promega) according to the manufacturer's instructions. Briefly, cells were seeded in white 96-well plates and treated for the indicated amount of time before adding AAF-Glo Reagent for 15 min. Luminescence was measured using a plate reader, before adding the lysis reagent and repeating the luminescence reading after 15 min of incubation. Blank samples were set up as wells containing a medium with no cells. Number of viable cells was calculated by subtracting the blank-normalized value of the first reading to that of the second reading, and expressed as a percent of the untreated samples.

## Measurement of NADt

Intracellular total NAD$^+$ and NADH levels were measured using the NAD/NADH-Glo assay (cat. no. G9071; Promega Corporation) following the supplier's instructions. Briefly, cells were seeded in white, flat-bottom 96-well plates (cat. no. EF86610K; Costar) the day before the experiment. The cells were treated for 6 h with equal amounts of MVs or left untreated. The cells were then incubated with 50 $\mu$l of NAD/NADH Glo detection reagent. After 1 h of incubation, luminescence was measured using a microplate reader.

## Western blotting

For Western blot analysis, equal amounts of proteins were separated using Novex WedgeWell 4–20%, Tris-Glycine gels (cat. no. XP04202BOX; Invitrogen, Thermo Fisher Scientific). Proteins were transferred onto PVDF membranes (cat. no. 88518; Thermo Fisher Scientific). Membranes were blocked with a solution of 5% non-fat dry milk (cat. no. 1706404XTU; Bio-Rad Laboratories) in Tris-buffered saline - 0.5% Tween-20 (TBS-T), then incubated with primary antibody in a solution of 5% BSA (cat. no. A2153; Bio-Rad Laboratories) in TBS-T overnight at 4°C. Membranes were then washed in TBS-T solution and incubated with HRP-conjugated secondary antibody (anti-mouse, cat. no. 7076, or anti-rabbit, cat. no. 7074; Cell Signalling Technology, Inc.) in 5% non-fat dry milk in TBS-T, for 1 h at room temperature. After additional washes, membranes were incubated with a chemiluminescent detection reagent (cat. no. NEL103E001EA; Western Lightning Plus, Chemiluminescent Substrate, PerkinElmer Inc.), and imaged on HyBlot CL Autoradiography Film (cat. no. 1159T41; Thomas Scientific), and the film was developed using developer and fixer solutions (Merry X-Ray Imaging, Inc.). The following primary antibodies were employed: p21 Waf1/Cip1 (cat. no. 37543S), HSP90 (cat. no. 4877), I$\kappa$B$\alpha$ (cat. no. 4812), NAMPT (cat. no. 6122), Poly/Mono-ADP Ribose (E6F6A) (cat. no. 83732), Acetyl-p53 (Lys379) (cat. no. 2570), $\beta$-actin (cat. no. 3700) and Flotillin-2 (cat. no. 3436), all from Cell Signaling Technology, Inc. Quantification of Western blots was performed using the ImageJ software (Schneider et al, 2012).

## Generation of GSC-267 NAMPTsh cells

A shRNA targeting NAMPT (sequence: GCTAGCAGCGATAGCTATGACAT TTATTACT-AGTATAAATGTCATAGCTATCGCTTTTTT) was selected using the Genetic Perturbation Platform, from the Broad Institute (https://www.broadinstitute.org/genetic-perturbation-platform). The shRNA oligo was obtained as a duplexed DNA from Integrated DNA Technologies, Inc., and cloned into the EZ-Tet-pLKO-Puro vector (Plasmid cat. no. 85966; AddGene) using the InFusion ligation kit (cat. no. 639650; Takara Bio). The cloned vector was sequenced to verify appropriate insertion of the shRNA. Lentiviruses were generated by transfecting HEK-293T cells with the shRNA plasmid, and the packaging plasmids (cat. no. 12259 and cat. no. 12263; AddGene) using Polyethylenimine (cat. no. 043896-03; Thermo Fisher Scientific). The viruses shed into the medium by the cells were harvested 24 and 48 h after transfection. GSC-267 cells were infected by treatment with the virus and polybrene (8 $\mu$g/ml). The following day, 1 $\mu$g/ml of puromycin was added to the media of infected GSC-267 cells to select for

cells carrying the construct, and selection medium was maintained for 8 d. NAMPT shRNA expression was induced by supplementing the medium with Dox 400 ng/ml. Dox was replenished every 48 h. GSC-267 NAMPT sh cells (both in the presence and absence of Dox) were maintained in the presence of NMN 125 $\mu$M to support normal cell metabolic activity and ability to generate MVs.

## NanoSight analysis

The size and concentration of EVs were measured using a NanoSight NS300 (Malvern, Cornell NanoScale Science and Technology Facility) as described previously (Kreger et al, 2016). Briefly, GSCs were grown in the absence of the B-27 supplement for 24 h before the conditioned medium was collected and centrifuged twice at 100$g$ for 5 min to pellet cells and debris. The partially clarified medium was then diluted in PBS and injected into the beam path to capture movies of EVs as points of diffracted light moving rapidly under Brownian motion. Five 45-s videos of each sample were taken and analyzed to obtain the concentration and size of the individual EVs based on their movement, and then the results were averaged.

# Data Availability

Data generated in this study are presented in the study and supporting files, and deposited as follows:

·MS-based proteomics data: deposited at the ProteomeXchange Consortium via the PRIDE partner repository (https://www.ebi.ac.uk/pride/archive/). Dataset identifier: PXD030092.

·Quantitative proteomic data of the CCLE: available in Nusinow et al (2020).

·Quantitative proteomic data of glioblastoma patients measured by the CPTAC: available in (Wang et al, 2021b).

·TCGA data: obtained through the https://www.cbioportal.org (Cerami et al, 2012; Gao et al, 2013), and through the Project Betastasis portal (https://www.betastasis.com).

# Supplementary Information

# Acknowledgements

We thank the Cornell Stem Cell and Transgenic Core Facility and especially Robert Munroe for their assistance with irradiating GSCs. This work was supported by NIH grant CA201402; Cornell Center for Vertebrate Genomics (CVG) Distinguished Scholar Award; the National Center for Research Resources (grant no. S10RR023781, providing the Mark I Model 68 Cesium-137 gamma irradiator), and by National Science Foundation grant NSF DMR-1719875.

## Author Contributions

E Panizza: conceptualization, data curation, formal analysis, supervision, funding acquisition, validation, investigation, visualization, methodology,

project administration, and writing—original draft, review, and editing.

BD Regalado: formal analysis, validation, investigation, and visualization.

F Wang: formal analysis and investigation.

I Nakano: conceptualization, supervision, funding acquisition, investigation, and writing—review and editing.

NM Vacanti: conceptualization, formal analysis, investigation, methodology, and writing—review and editing.

RA Cerione: conceptualization, resources, supervision, funding acquisition, investigation, project administration, and writing—review and editing.

MA Antonyak: conceptualization, resources, supervision, funding acquisition, investigation, project administration, and writing—original draft, review, and editing.

## Conflict of Interest Statement

The authors declare that they have no conflict of interest.

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
