## [Reviewer comments · Life Science Alliance]

Life Science Alliance

Proteomic analysis reveals microvesicles containing NAMPT as mediators of radio-resistance in glioma

Elena Panizza, Brandon Regalado, Fangyu Wang, Ichiro Nakano, Nathaniel Vacanti, Richard Cerione, and Marc Antonyak
DOI: <https://doi.org/10.26508/lsa.202201680>

Corresponding author(s): Richard Cerione, Cornell University and Marc Antonyak, Cornell University

Review Timeline:

Submission Date:	2022-08-18
Editorial Decision:	2022-10-24
Revision Received:	2023-02-23
Editorial Decision:	2023-03-23
Revision Received:	2023-03-29
Accepted:	2023-03-29

Transaction Report:

October 24, 2022

Re: Life Science Alliance manuscript #LSA-2022-01680-T

Dr. Richard A. Cerione
Dept. of Molecular Medicine
College of Veterinary Medicine
Cornell University
Ithaca, Veterinary Medical Center C3-155
Ithaca, NY 14853-6401

Dear Dr. Cerione,

Thank you for submitting your manuscript entitled "Proteomic analysis reveals microvesicles containing NAMPT as mediators of radiation resistance in glioma" to Life Science Alliance. The manuscript was assessed by expert reviewers, whose comments are appended to this letter. We invite you to submit a revised manuscript addressing the Reviewer comments.

Thank you for this interesting contribution to Life Science Alliance. We are looking forward to receiving your revised manuscript.

Sincerely,

B. MANUSCRIPT ORGANIZATION AND FORMATTING:

Reviewer #1 (Comments to the Authors (Required)):

The authors show that the MV-97 mediated transfer of NAMPT increases the total NAD(H) level in recipient cells, and promotes their proliferation upon irradiation, exposure to low serum, or treatment with the radiation mimetic Bleomycin. These effects occur in fibroblasts as well as in radio-sensitive GSCs that were treated with NAMPT-high MVs. They further show that NAMPT transfer and its enzymatic activity in MV-recipient cells are both required to promote the radio-resistant phenotype. These findings highlight how some NAMPT-overexpressing GSCs and glioma cells promote resistance to radiation by modulating their surroundings through the shedding and transfer of NAMPT-high MVs. They also raise the interesting possibility that strategies to block the production of MVs, or to intervene against the increased NAD(H) levels (and its consequences), can potentially be combined with radiation to more effectively treat glioma patients that overexpress NAMPT.

The work is to some extent well done and deserves publication.

However, it can be improved, especially conclusions, incorporating some further experiments.

- NAMPT exist and is functional at extracellular level (visfatin). Does this account for the effect observed?. For example, add visfatin (extracellular nampt) to the normal media of radiosensitive cells and measure acquired new resistance (if so).
- Add conditioned media but eliminating microvesicles and measure again whether the effect is produced, by soluble visfatin or it needs the presence of microvesicles?
- It has been reported that NAD itself can induce resistance to apoptosis. Can the adding of NAD reproduce this effect of EV enriched in NAMPT?
- To some extent can this be reproduced in vivo? Orthotopic models will be better, but xenograft will be enough for this first approx.

Reviewer #2 (Comments to the Authors (Required)):

Radiation remains the standard of care for glioblastoma tumors, and relatively few targeted strategies for treating these tumors have been identified, despite the well-known fact that most tumors will recur after irradiation. Here, the authors use a suite of patient-derived glioma stem cell cultures (GSC lines) to study protein level changes pre and post irradiation in vitro. They then focus on extracellular vesicles, specifically microvesicles derived from one of the GSC lines with enhanced resistance to radiation (defined as no change in relative proliferation after irradiation), and find that these microvesicles can confer radiation resistance to cells that are normally sensitive. This resistance is achieved through transfer of the NAMPT enzyme and presumed increase of cellular NAD(H) levels, suggesting a mechanism by which radiation-resistant subclones within heterogeneous patient tumors could transfer resistance to neighboring cells.

This work provides a new and interesting complement to existing proteomic datasets on glioblastoma cell lines and patient samples, in particular through the focus on extracellular vesicles and transfer of enzymatic activity. The study could be strengthened further by additional exploration of the downstream consequences of NAMPT activity, and in some cases the language interpreting data should be more measured.

Major points:

- 1) The initial section of the paper focuses on distinguishing radioresistant and radiosensitive GSC lines based on significant protein changes upon radiation - however, the majority of the subsequent experiments then use MVs from unirradiated cells. Is NAMPT further increased in GSC267 cells (or any other lines) following irradiation? Or is it continuously present at similar levels irrespective of treatment? Please clarify.
- 2) The cell proliferation assay used in multiple figures (CCK8 assay kit) appears to be NADH/NAD⁺ dependent in its mechanism. Could these proliferation data be reflecting changes in this pathway's activity levels but not necessarily proliferation rate or viability? Did the authors confirm using a complementary approach (e.g. cleaved caspase-3 staining, other viability dyes, hand counting for apoptosis, or doubling time assays, nucleoside analog incorporation for proliferation)? A second plate-based counting assay is listed in methods, but it is not clear when this assay and/or CCK8 were used based on the labeling and figure legends. Some additional quantification and verification in this aspect would strengthen confidence in the conclusions of the

paper.

3) How durable are the effects of microvesicle transfer of NAMPT? Or, put another way, how long after a single MV treatment do the cells retain radiation resistance?

4) In the introduction, the authors highlight at least two ways in which cellular NAD(H) regulation could enable radiation resistance. It would strengthen the conclusions to determine if either of these mechanisms (regulation of p53 or PARP) are active in their experimental system, for example through a Western blot for acetylated p53 or its downstream targets.

5) In some spots, the language interpreting the results should be tempered - while the conclusions advanced are certainly possible explanations for what is seen, rigorous statistical testing needs to be included to support the claims as written. For example - the authors indicate, in line 142 and elsewhere, that "IDH mutant/hypermethylated status.....[is not] associated with the GSCs radio-sensitivity status. But, as shown, all three IDHmut lines do cluster together in the radio-sensitive group - and the study does not appear sufficiently powered to test whether such an association is or is not present.

Similarly, lines 83-85 say that "we identify distinct oncogenic drivers across the resistant GSC lines..." - but only NAMPT is tested to a level sufficient to meet this claim. This sentence should be tempered - presence / overexpression of particular proteins does not necessarily mean that they are the drivers or mechanisms of resistance, and it's not clear from the data shown whether NAMPT production is specific (for example) to the mesenchymal subtype.

Minor points:

1) Line 49 - please include additional citations of relevant work for the "first existence of glioma stem cells" - there are many groups that have done complementary work in this area (for example, Singh et al doi: 10.1038/nature03128, which precedes the currently cited work by two years!).

2) Lines 71-72: Citations and clarification needed here - are the authors referring to experiments in vitro here? Presumably yes but it reads as if there are 2 cancers of different grade in one individual.

3) Have the GSC lines described here been deposited with a public repository such as the CPDM at Dana-Farber? If no, will the underlying genomic data classifying the lines be made available, consistent with NIH data sharing standards? These data are not mentioned in the availability statement.

4) Figure 1D - what do the individual data points represent? Separate GSC lines? Technical replicates?

5) Please quantify the Western blot shown in Figure 1 - figure supplement 1. The differences between radio-sensitive / radio-resistant cells in induction of p21/CDKN1A are not readily apparent by eye and don't look comparable to Figure 1D.

Referee Cross-Comments: While I agree that an orthotopic or other model would be a useful extension this type of experiment may be beyond the scope of the article as it stands. Perhaps another approach would be to visualize message and protein for NAMPT in human tumor sections - presumably some cells should be producing both but others would only have evidence of the protein, if the model of MV transfer is correct.

Reviewer #1 (Comments to the Authors (Required)):

The authors show that the MV-267 mediated transfer of NAMPT increases the total NAD(H) level in recipient cells, and promotes their proliferation upon irradiation, exposure to low serum, or treatment with the radiation mimetic Bleomycin. These effects occur in fibroblasts as well as in radio-sensitive GSCs that were treated with NAMPT-high MVs. They further show that NAMPT transfer and its enzymatic activity in MV-recipient cells are both required to promote the radio-resistant phenotype. These findings highlight how some NAMPT-overexpressing GSCs and glioma cells promote resistance to radiation by modulating their surroundings through the shedding and transfer of NAMPT-high MVs. They also raise the interesting possibility that strategies to block the production of MVs, or to intervene against the increased NAD(H) levels (and its consequences), can potentially be combined with radiation to more effectively treat glioma patients that overexpress NAMPT.

The work is to some extent well done and deserves publication. However, it can be improved, especially conclusions, incorporating some further experiments.

1 - NAMPT exists and is functional at extracellular level (visfatin). Does this account for the effect observed? For example, add visfatin (extracellular nampt) to the normal media of radiosensitive cells and measure acquired new resistance (if so).

Our results based on experiments performed on NIH-3T3 cells show that extracellular NAMPT (human recombinant NAMPT, hrNAMPT) does not confer resistance to radio-sensitive cells (Fig 4H). To further substantiate this finding, a similar experiment was carried out using GSC-408 cells as the radio-sensitive cell model. Again, we determined that adding hrNAMPT to these cells does not confer radiation resistance (Fig 4G).

2 - Add conditioned media but eliminating microvesicles and measure again whether the effect is produced, by soluble visfatin or it needs the presence of microvesicles?

To answer this question, we treated radio-sensitive cells GSC-408 with different combinations of irradiation and conditioned medium derived from GSC-267 cells that had been depleted of MVs. The addition of the MV-depleted conditioned media to the cells did not promote their proliferation following irradiation. These new findings show that the MV fraction is critical for mediating this effect and have now been included in supplementary Fig S2C.

3 - It has been reported that NAD itself can induce resistance to apoptosis. Can the adding of NAD reproduce this effect of EV enriched in NAMPT?

To address this question, we treated cells with the cell permeable NAD⁺ precursor nicotinamide mononucleotide (NMN), which is readily converted into NAD intracellularly. The addition of

NMN does confer a similar proliferative advantage to fibroblasts as that observed when treating cells with MVs enriched in NAMPT (i.e., MVs derived from GSC-267 cells; see Fig 4H). Furthermore, an additional experiment was performed to determine whether treating irradiated GSC-408 cells (a radio-sensitive cell line) with NMN can also increase their proliferation. Our new results show that the addition of NMN to these cells does indeed enhance their proliferation (Fig 4G).

4 - To some extent can this be reproduced in vivo? Orthotopic models will be better, but xenograft will be enough for this first approx.

Please see our response to the referee cross-comment at the end of the point-by-point for a detailed answer to this question.

Reviewer #2 (Comments to the Authors (Required)):

Radiation remains the standard of care for glioblastoma tumors, and relatively few targeted strategies for treating these tumors have been identified, despite the well-known fact that most tumors will recur after irradiation. Here, the authors use a suite of patient-derived glioma stem cell cultures (GSC lines) to study protein level changes pre and post irradiation in vitro. They then focus on extracellular vesicles, specifically microvesicles derived from one of the GSC lines with enhanced resistance to radiation (defined as no change in relative proliferation after irradiation), and find that these microvesicles can confer radiation resistance to cells that are normally sensitive. This resistance is achieved through transfer of the NAMPT enzyme and presumed increase of cellular NAD(H) levels, suggesting a mechanism by which radiation-resistant subclones within heterogeneous patient tumors could transfer resistance to neighboring cells.

This work provides a new and interesting complement to existing proteomic datasets on glioblastoma cell lines and patient samples, in particular through the focus on extracellular vesicles and transfer of enzymatic activity. The study could be strengthened further by additional exploration of the downstream consequences of NAMPT activity, and in some cases the language interpreting data should be more measured.

Major points:

1) The initial section of the paper focuses on distinguishing radioresistant and radiosensitive GSC lines based on significant protein changes upon radiation - however, the majority of the subsequent experiments then use MVs from unirradiated cells. Is NAMPT further increased in GSC267 cells (or any other lines) following irradiation? Or is it continuously present at similar levels irrespective of treatment? Please clarify.

NAMPT expression levels do not change in either radio-sensitive or radio-resistant GSCs upon radiation treatment. This result was included in supplementary Fig S3B.

2) *The cell proliferation assay used in multiple figures (CCK8 assay kit) appears to be NADH/NAD⁺ dependent in its mechanism. Could these proliferation data be reflecting changes in this pathway's activity levels but not necessarily proliferation rate or viability? Did the authors confirm using a complementary approach (e.g. cleaved caspase-3 staining, other viability dyes, hand counting for apoptosis, or doubling time assays, nucleoside analog incorporation for proliferation)? A second plate-based counting assay is listed in methods, but it is not clear when this assay and/or CCK8 were used based on the labeling and figure legends. Some additional quantification and verification in this aspect would strengthen confidence in the conclusions of the paper.*

We thank the reviewer for this comment, and for pointing out the need to further clarify how cell proliferation was determined in our experiments. All plots with y-axes labelled “relative cell no.” (Fig 3G, 4F, and supplementary Fig S4D-E) involved counting cells that were stained with Hoechst and propidium iodide to determine the total number of cells and dead cells, respectively. These experimental details are included in *Methods*. Moreover, representative fluorescent microscopy images of the cell counting results plotted in Fig 4F were included as supplementary Fig S4C.

To measure the proliferation of radio-sensitive and radio-resistant GSCs upon radiation using an additional approach, we performed viable cell counting based on the quantification of intracellular proteases (CytoTox Glo assay, see the *Methods* section). Our results show that the number of viable radiation-resistant GSC-267 cells four days after being irradiated is comparable to that of untreated cells, whereas the number of irradiated GSC-408 cells (a radiation-sensitive cell line) was reduced to about 60% of the control (supplementary Fig S1D). These findings are similar to those obtained when we used the CCK8 assay kit in Fig 1C.

3) *How durable are the effects of microvesicle transfer of NAMPT? Or, put another way, how long after a single MV treatment do the cells retain radiation resistance?*

This is an interesting question that has been the matter of debate in the extracellular vesicle (EV) field. Within the tumor microenvironment, EVs are continuously being produced by aggressive cancer cells and are transferred to less aggressive and non-cancerous cell types that comprise the microenvironment. Therefore, it is reasonable to speculate that regularly treating recipient (i.e., target) cells with EVs is a good way of mimicking the conditions occurring within the tumor microenvironment. In line with this idea, our previous work (Antonyak et al., PNAS 2011) shows that colony formation in soft agar by NIH/3T3 cells is dependent on their repeated exposure to MVs derived from cancer cells.

To determine whether the continuous exposure to MVs enriched in NAMPT was necessary to maintain resistance traits in recipient cells, we treated radio-sensitive GSC-408 cells with MVs for four days, then allowed them to grow for four additional days in regular medium before irradiation. The proliferation of the pre-conditioned GSC-408 cells was reduced by about 50%

after irradiation compared to control cells, similar to what was observed for GSC-408 cells that had not been pre-conditioned with MVs (supplementary Fig S3E). Thus, our results suggest that continuous exposure to MVs containing NAMPT is necessary to confer radiation resistance to sensitive cells.

4) In the introduction, the authors highlight at least two ways in which cellular NAD(H) regulation could enable radiation resistance. It would strengthen the conclusions to determine if either of these mechanisms (regulation of p53 or PARP) are active in their experimental system, for example through a Western blot for acetylated p53 or its downstream targets.

In order to address the point raised by the reviewer, we measured the levels of acetylation of p53 at lysine 382 (K382), mono-poly ADP ribosylation and p21 expression, in sensitive GSC-408 cells and resistant GSC-267 cells upon radiation. The resulting western blots have been included in Fig 2I-J and are described in the corresponding Results section as follows:

“...radiation treatment results in the rapid acetylation of p53 at lysine residue 382 (K382) in NAMPT-low GSC-408 cells, but not in NAMPT-high GSC-267 cells. Accordingly, the expression of the p53 transcriptional target p21, which directs cell cycle arrest upon DNA damage, is not detected in GSC-267 cells, while it is increased in GSC-408 cells upon radiation (**Fig 2I-J**). NAD⁺ is also necessary for the function of PARPs, which initiate DNA repair by adding poly/mono ADP ribose chains (PARylation) at break sites (Kennedy et al., 2016). Indeed, PARylation is strongly activated in GSC-267 cells within 2.5-5 hours after irradiation, while it is not in GSC-408 cells (**Fig 2I-J**)”

5) In some spots, the language interpreting the results should be tempered - while the conclusions advanced are certainly possible explanations for what is seen, rigorous statistical testing needs to be included to support the claims as written. For example - the authors indicate, in line 142 and elsewhere, that "IDH mutant/hypermethylated status.....[is not] associated with the GSCs radio-sensitivity status. But, as shown, all three IDHmut lines do cluster together in the radio-sensitive group - and the study does not appear sufficiently powered to test whether such an association is or is not present.

We thank the reviewer for the comment. We have rephrased the results in line 142 as follows: “We further establish that the GSC transcriptional subtype (Mao et al., 2013) is not associated with their radio-sensitivity status (**Fig 1A**). Similarly, we find that IDH mutant negative GSCs (Baysan et al., 2012; Wang et al., 2017) (**supplementary Fig S1E**) can be either sensitive or resistant to radiation (**Fig 1A**).”

Similarly, lines 83-85 say that "we identify distinct oncogenic drivers across the resistant GSC lines..." - but only NAMPT is tested to a level sufficient to meet this claim. This sentence should be tempered - presence / overexpression of particular proteins does not necessarily mean that they are the drivers or mechanisms of resistance, and it's not clear from the data shown whether NAMPT production is specific (for example) to the mesenchymal subtype.

We have also rephrased the above-mentioned sentence as follows to better reflect the results shown:

“Additionally, we identify distinct oncogenic driver proteins that are overexpressed across the resistant GSC lines, suggesting that different disease mechanisms exist among the patients from which the cell lines were derived. “

Specifically, our results indicate overexpression of proteins that are well documented oncogenic drivers in different GSC lines. Overexpression of such oncogenic driver proteins indicates a role in driving disease, as detailed in previous publication (Lehtiö et al., Nature Cancer 2021).

Minor points:

1) *Line 49 - please include additional citations of relevant work for the "first existence of glioma stem cells" - there are many groups that have done complementary work in this area (for example, Singh et al doi: 10.1038/nature03128, which precedes the currently cited work by two years!).*

We included the indicated reference and amended the text in the Results as follows:

“In 2004, the existence of glioma stem cells (GSCs) was first reported”.

2) *Lines 71-72: Citations and clarification needed here - are the authors referring to experiments in vitro here? Presumably yes but it reads as if there are 2 cancers of different grade in one individual.*

The sentence was clarified, and citations included as follows:

“EVs produced by cancer cells are transferred to other cells within the tumor microenvironment, which may include non-transformed cells and less aggressive cancer cells, and significantly promote their ability to proliferate, as well as exhibit therapy resistance and invasiveness (Burgos-Ravanel et al., 2021; Maacha et al., 2019; Xu et al., 2018)”.

3) *Have the GSC lines described here been deposited with a public repository such as the CPDM at Dana-Farber? If no, will the underlying genomic data classifying the lines be made available, consistent with NIH data sharing standards? These data are not mentioned in the availability statement.*

The GSC lines described in this manuscript have not been deposited in a public repository. However, the cell lines have been extensively characterized and studied in several publications, including being subjected to mRNA profiling and immunostaining (see below for details). Here, we further characterized these GSCs, based on their expression of signatures for the different glioma subtypes. This information has now been included in *Methods* to highlight the studies where the GSCs were used:

“Cell culture and treatment

GSC lines were characterized based on mRNA profiling and/or immunostaining, as described upon their publications (Guvenc et al., 2013, Mao et al., 2013, Kim et al., 2016, Adnani et al., 2022, Alhalabi et al., 2022). GSC identity was also confirmed based on their expression of signatures corresponding to mesenchymal, classical or proneural subtypes (Verhaak et al., 2010), as detailed in **supplementary Fig S1C.**"

4) Fig 1D - what do the individual data points represent? Separate GSC lines? Technical replicates?

The individual points in Fig 1D represent cell lines. This has been clarified in the respective figure legend.

5) Please quantify the Western blot shown in Fig 1 - figure supplement 1. The differences between radio-sensitive / radio-resistant cells in induction of p21/CDKN1A are not readily apparent by eye and don't look comparable to Fig 1D.

Both cell lines analyzed in the western blot shown in supplementary Fig S1B (GSC-1079 and GSC-374) are identified as radio-sensitive in later analyses. Accordingly, they both display a strong activation of p21 in response to ionizing radiation.

Referee Cross-Comments: While I agree that an orthotopic or other model would be a useful extension this type of experiment may be beyond the scope of the article as it stands. Perhaps another approach would be to visualize message and protein for NAMPT in human tumor sections - presumably some cells should be producing both but others would only have evidence of the protein, if the model of MV transfer is correct.

One difficulty with performing the analysis suggested by the reviewer is that NAMPT is ubiquitously expressed by all cell types as it is an essential metabolic protein. Therefore, the task would be that of capturing differences between the levels of NAMPT protein and that of its mRNA, which could be small and thus difficult to detect. Additionally, we could not find any data set where NAMPT mRNA and protein were quantified at a single cell level in tumor sections.

March 23, 2023

RE: Life Science Alliance Manuscript #LSA-2022-01680-TR

Dr. Richard A. Cerione
Cornell University
Dept. of Molecular Medicine
College of Veterinary Medicine
Cornell University
Ithaca, Veterinary Medical Center C3-155 USA-Ithaca, NY 14853-6401

Dear Dr. Cerione,

Thank you for submitting your revised manuscript entitled "Proteomic analysis reveals microvesicles containing NAMPT as mediators of radio-resistance in glioma". We would be happy to publish your paper in Life Science Alliance pending final revisions necessary to meet our formatting guidelines.

- please address Reviewer 2's remaining comment
- please add ORCID ID for both corresponding authors; you should have received instructions on how to do so
- please use the [10 author names, et al.] format in your references (i.e. limit the author names to the first 10)

Figure Check:

- please add sizes next to all blots
- in Figure 3F: what are the black vertical lines meant to indicate?

A. FINAL FILES:

B. MANUSCRIPT ORGANIZATION AND FORMATTING:

Sincerely,

Reviewer #2 (Comments to the Authors (Required)):

The authors have largely answered my concerns, and publication is appropriate.

One remaining point from the cross comment that went unaddressed is staining for NAMPT protein in tumor sections - as the authors point out, it might be challenging to compare relative transcript levels and relative protein levels in the same section. However, based on the ideas advanced in the article, it should be straightforward to detect variable cell-to-cell levels of protein via standard staining techniques. Recommend a minor revision to discussion suggesting that this could be studied in future.

- please address Reviewer 2's remaining comment

Reviewer #2 (Comments to the Authors (Required)):

The authors have largely answered my concerns, and publication is appropriate.

One remaining point from the cross comment that went unaddressed is staining for NAMPT protein in tumor sections - as the authors point out, it might be challenging to compare relative transcript levels and relative protein levels in the same section. However, based on the ideas advanced in the article, it should be straightforward to detect variable cell-to-cell levels of protein via standard staining techniques. Recommend a minor revision to discussion suggesting that this could be studied in future.

To address this comment, we included the following text in the Discussion section on page 10 of the manuscript file:

“To extent to which the EV-mediated transfer of NAMPT occurs in the tumor microenvironment remains to be determined, and a possible way to examine it will be to quantify differences in NAMPT protein expression levels across single cells within tumor sections. This is an important question that will be worth addressing in future studies.”

- in Figure 3F: what are the black vertical lines meant to indicate?

The black vertical lines indicated that the WB portions depicted on the figure were not adjacent on the original blot. However, upon revision of the figure adjacent portions were used instead and the vertical lines were removed.

March 29, 2023

RE: Life Science Alliance Manuscript #LSA-2022-01680-TRR

Dr. Richard A. Cerione
Cornell University
Dept. of Molecular Medicine
College of Veterinary Medicine
Cornell University
Ithaca, Veterinary Medical Center C3-155 USA-Ithaca, NY 14853-6401

Dear Dr. Cerione,

Thank you for submitting your Research Article entitled "Proteomic analysis reveals microvesicles containing NAMPT as mediators of radio-resistance in glioma". It is a pleasure to let you know that your manuscript is now accepted for publication in Life Science Alliance. Congratulations on this interesting work.

DISTRIBUTION OF MATERIALS:

Again, congratulations on a very nice paper. I hope you found the review process to be constructive and are pleased with how the manuscript was handled editorially. We look forward to future exciting submissions from your lab.

Sincerely,
